# Echocardiography for the Assessment of Pulmonary Hypertension and Congenital Heart Disease in the Young

**DOI:** 10.3390/diagnostics11010049

**Published:** 2020-12-31

**Authors:** Katharina Meinel, Martin Koestenberger, Hannes Sallmon, Georg Hansmann, Guido E. Pieles

**Affiliations:** 1Division of Pediatric Cardiology, Department of Pediatrics, Medical University Graz, 8036 Graz, Austria; katharina.meinel@medunigraz.at (K.M.); martin.koestenberger@medunigraz.at (M.K.); 2European Pediatric Pulmonary Vascular Disease Network, 13125 Berlin, Germany; Hannes.Sallmon@charite.de (H.S.); georg.hansmann@gmail.com (G.H.); 3Department of Pediatric Cardiology, Charité-Universitätsmedizin Berlin, 13353 Berlin, Germany; 4Department of Congenital Heart Disease/Pediatric Cardiology, Deutsches Herzzentrum Berlin (DHZB), 13353 Berlin, Germany; 5Department of Pediatric Cardiology and Critical Care, Hannover Medical School, 30625 Hannover, Germany; 6National Institute for Health Research (NIHR) Cardiovascular Biomedical Research Centre, Congenital Heart Unit, Bristol Royal Hospital for Children and Bristol Heart Institute, Bristol BS2 8HW, UK; 7Institute of Sport Exercise and Health (ISEH), University College London, London W1T 7HA, UK

**Keywords:** pulmonary hypertension, pulmonary arterial hypertension, congenital heart disease, echocardiography, exercise assessment

## Abstract

While invasive assessment of hemodynamics and testing of acute vasoreactivity in the catheterization laboratory is the gold standard for diagnosing pulmonary hypertension (PH) and pulmonary vascular disease (PVD) in children, transthoracic echocardiography (TTE) serves as the initial diagnostic tool. International guidelines suggest several key echocardiographic variables and indices for the screening studies when PH is suspected. However, due to the complex anatomy and special physiological considerations, these may not apply to patients with congenital heart disease (CHD). Misinterpretation of TTE variables can lead to delayed diagnosis and therapy, with fatal consequences, or–on the other hand-unnecessary invasive diagnostic procedures that have relevant risks, especially in the pediatric age group. We herein provide an overview of the echocardiographic workup of children and adolescents with PH with a special focus on children with CHD, such as ventricular/atrial septal defects, tetralogy of Fallot or univentricular physiology. In addition, we address the use of echocardiography as a tool to assess eligibility for exercise and sports, a major determinant of quality of life and outcome in patients with PH associated with CHD.

## 1. Introduction

Pulmonary hypertension (PH) associated with congenital heart disease (CHD) is a complex condition associated with increased morbidity and mortality that can affect individuals at any age [1,2,3]. According to the 6th World Symposium on PH (WSPH) 2018, PH is currently defined as a mean pulmonary artery pressure (mPAP) > 20 mm Hg measured by cardiac catheterization at rest [1,2]. For consistency, this threshold was also used for biventricular circulations in the latest pediatric consensus statements [1,2,3]. Patients with PH-CHD can be classified according to the guidelines, as shown in Table 1 and Table 2 [1,2].

The term pulmonary arterial hypertension (PAH) (group 1 PH) is hemodynamically defined as precapillary PH [1,2] and characterized by pulmonary vascular remodeling leading to increased pulmonary vascular resistance (PVR) and ultimately to pulmonary vascular disease (PVD) and right heart failure [4,5]. In contrast to idiopathic PAH (IPAH) and hereditable PAH (HPAH), PAH associated with CHD (PAH-CHD) (group 1.4.4 PH or group 5.4 PH = complex CHD, Table 2) is common in children [2,6] and frequently occurs with post-tricuspid left-to-right cardiovascular shunts, ultimately leading to PVR elevation. It should be noted that the definition of PAH requires a PVR index ≥ 3 WU x m^2^, in addition to mPAP > 20 mm Hg and PAWP or LVEDP ≤ 15 mm Hg. Thus, PAH with only slightly elevated PVR index is neither synonymous nor indicative of PVD in patients with CHD [7,8]. Systemic-to-pulmonary shunts, for instance, may lead to an enhanced pulmonary blood flow and consequently to an increased mPAP with or without PVD, as PVR (in children: PVR indexed [PVRi]) may be within the normal range [7,9].

Therefore, this differentiation is especially important when assessing PAH-CHD patients for operability as children without PVD may benefit from closure of left-to-right shunts, whereas children with PAH-CHD and (significant) PVD may do not tolerate shunt repair [8,9]. Depending on whether the shunt is localized pre or post the tricuspid valve, the onset of progression of PVD in PAH-CHD varies greatly. Large post-tricuspid left-to-right shunts, leading to both volume and pressure overload of the pulmonary circulation (e.g., ventricular septal defects (VSD) or persistent ductus arteriosus [PDA]), progress more frequently and earlier to PVD and Eisenmenger syndrome than pre-tricuspid shunt lesions with low-pressure levels (e.g., atrial septal defect [ASD]), which are leading to volume overload of the RV and the pulmonary circulation [9,10,11]. However, PAH and/or PVD may persist, recur or even develop with a prevalence of 2–6% in children (5–13% in adults) despite complete surgical repair of CHD [6,12,13]. Those findings are particularly seen after repair of conotruncal lesions (e.g., tetralogy of Fallot (TOF), pulmonary stenosis, and dextro-transposition of the great arteries (d-TGA), among others) [8,14], and in patients with genetic abnormalities such as trisomy 21 (Down’s syndrome).

Other types of precapillary PH in the context of CHD include PH associated with lung disease (group 3 PH), PH due to pulmonary artery (PA) obstruction (group 4 PH, Table 2) and PH due to complex CHD (group 5.4 PH, Table 2) including, e.g., segmental PH, scimitar syndrome or Fontan physiology [1]. In the latter group, diagnosis and management of PH/PAH are particularly challenging as patients with Fontan physiology, for instance, do not fulfill standard criteria of PH as PVR(i) may be increased, despite an mPAP < 20 mm Hg (Table 1) [1,2].

PH associated with left heart disease (group 2.4, Table 2), hemodynamically characterized as postcapillary PH, may occur in children as a result of congenital or acquired left heart inflow and/or outflow obstructions (e.g., pulmonary vein stenosis, mitral/aortic stenosis, coarctation of the aorta) and systolic and/or diastolic systemic ventricular dysfunction [2]. Postcapillary PH secondary to left heart disease can be isolated or (quite frequently) combined with a precapillary PH component (Table 1) [1,2]. The details and caveats of the hemodynamic measurements in the catheterization laboratory, including the need to perform right and left heart catheterization, especially in PAH-CHD and in pediatric and adult patients with a suspected postcapillary component of PH, is discussed elsewhere [15].

Physiologically, there are three components to the RA function, including the reservoir-, conduit- and pump function. In states of RV diastolic dysfunction, right atrial (RA) contractility increases, and the RA becomes more distensible to maintain RV filling [16,17]. However, this compensatory response is leading to an increase of RA pressure (RAP) and RA size over time until the RA can no longer generate sufficient preload [18]. In the context of PH, elevated RAP is known to be a risk factor for mortality, and an increased RA size is prognostic of adverse outcomes in adults and children [19,20,21]. However, in our experience, in children, even in those with suprasystemic PA pressure, mRAP and RVEDP are rarely severely elevated > 15 mm Hg and often well below 15 mm Hg even in endstage disease. Similarly, an impaired left ventricular (LV) relaxation leads to an increased LA pressure, congestion of the pulmonary vessels and to chronic pulmonary vascular changes [22]. In children and adults with Fontan physiology, a raised diastolic pressure of the systemic ventricle reduces cardiac filling and worsens systemic venous hypertension [23]. In biventricular circulations, RV pressure/volume overload, such as in PH with or without CHD, a significantly dilated hypertensive RV undergoes RV-PA uncoupling when RV contractility does meet the increased pressure afterload. To the subsequent insufficient RV, output leads to inadequate pulmonary blood flow and, as such insufficient LV preload supply, entailing reduced LV filling and LV stiffness [24,25,26]. This mechanism is additionally reinforced through direct systolic compression of the LV by the RV and the leftward flattening of the interventricular septum (IVS) [27]. Particularly for follow-up and treatment of PH-CHD patients, it is important to differentiate between RV volume and pressure overload, as the LV ejection fraction (LVEF) is typically preserved in states of RV pressure overload, whereas LVEF is frequently decreased in RV volume overloading despite an increased LV end-diastolic volume [28].

A clinical hallmark of progressive PH is fading physical activity [29], highlighted by a decreased exercise capacity (VO2 peak) [30], a decline in quality of life, and a grim prognosis [4,31,32]. Exercise testing in the majority of pediatric PH patients is feasible and safe [33]. It is important to realize that besides metabolic and respiratory parameters, RV function is one of the determining factors of exercise capacity and causative in the pathophysiology of the often debilitating exercise intolerance and social isolation experienced by patients suffering from PH/PAH and RV dysfunction [32]. Furthermore, RV dysfunction is correlated to exercise performance and in turn to clinical outcome [34], and accurate assessment of RV function during exercise could have a role in estimating and monitoring exercise performance in children with PH at baseline and after recently introduced rehabilitation programs for children with PH [35]. Along these lines, it is crucial to note for PH healthcare providers that diastolic PAP and diastolic transpulmonary pressure gradient (dTPG, syn. DPG) is largely independent of RV stroke volume (while sPAP and mPAP is not), and thus–in the absence of significant pulmonary valve regurgitation, a surrogate of PVR and PVD at best directly measured in the catheterization laboratory.

Although the definite diagnosis of PH should be confirmed by cardiac catheterization, TTE plays a key role in identifying patients with signs of increased mPAP and is the primary noninvasive modality of choice to describe morphology, hemodynamics, ventricular systolic/diastolic function and ventricular–ventricular interactions (VVI) [36,37,38]. This manuscript provides an update on echocardiographic indicators of cardiac performance beyond commonly used variables to assess, e.g., RV systolic function, such as the tricuspid annular peak systolic excursion (TAPSE) with a special focus on PH/PAH-CHD (Table 3). Timely clinical assessment of disease progression in PAH-CHD, operability, follow-up and initiation of targeted PAH-pharmacotherapy is essential in order to prevent PVD [3,9,39]. The additional implementation of cardiac MRI or chest CT for exact determination of ventricular volumes, obstructions or valvular regurgitation is recommended in the presence of suboptimal echocardiographic windows or complex CHD [40]. Both cardiac MRI and chest CT are also recommended for the initial assessment of patients with PH at diagnosis, in addition to echocardiography and cardiac catheterization [3].

To assess risk in pediatric PH, in 2019, the European Pediatric Pulmonary Vascular Disease Network (EPPVDN) established a novel risk score [3], which was recently applied in a prospective analysis in children with PAH, including PAH-CHD patients, under double or triple oral therapy [41]. The EPPVDN risk score has been shown, be it combined invasive/noninvasive or noninvasive only, to reliably indicate a change of clinical status with medication and to reliably determine the risk when compared with established single determinants of risk and outcome in children with PH [3,41,42].

### 1.1. Echocardiographic Features of the RV in PH-CHD

#### Estimation of Systolic PAP (sPAP)

The estimation of sPAP is based on the peak tricuspid regurgitation velocity (TRV) (Figure 1B). The simplified Bernoulli equation (sPAP = 4 × v^2^ + mean RAP [right atrial pressure]), using continuous wave (CW)-Doppler to assess the TRV, describes the relationship of TR and right ventricular systolic pressure (RVSP) as a surrogate of sPAP [43]. RAP can be estimated on the basis of the diameter, collapsibility and distensibility of the inferior vena cava (IVC). However, in children, RAP values between 5–10 mm Hg are usually assumed in clinical practice [37]. For estimating sPAP correctly, other causes for elevated RAP, e.g., ASD, tricuspid stenosis or restrictive RV physiology, must be excluded [44]. A TRV > 3.4 m/s, corresponding to a sPAP >50 mm Hg at rest, makes PH highly likely [45,46].

However, several limitations should be considered: The application of the TRV for estimating sPAP is only permitted in the absence of right ventricular outflow tract (RVOT) obstruction or valvular/supravalvular PA stenosis [43]. As TRV does not directly reflect sPAP under these conditions, but rather the RVSP, evaluation of the forward pulmonary valve gradient in conjunction with the TR gradient is advisable [43,47]. Furthermore, RVSP and, therefore, sPAP may be underestimated in cases with torrential TRV and suboptimal Doppler spectral envelopes [37]. Of note, in patients with Fontan physiology, the regurgitation of the atrioventricular valve(s) (AVV) provides no information on sPAP as there is no sub-pulmonary ventricle [47].

In the presence of a VSD, sPAP can be estimated through measurement of the maximal velocity across the VSD by CW-Doppler in the absence of RVOT and LV outflow tract (LVOT) obstructions [48]. In VSDs with left-to-right shunts, the obtained peak Doppler gradient needs to be subtracted from systolic blood pressure (SBP), whereas in right-to-left shunting, it needs to be added to SBP [49,50]. By using this method, good correlations with invasively obtained measurements by cardiac catheterization have been shown, in which low velocities across the VSD as well as right-to-left shunting likely indicate PH [48,51]. Similarly, in systemic to pulmonary shunts (e.g., PDA or surgical Shunts like Blalock–Taussig (BT) and Potts shunt), the peak Doppler gradient across the shunt may allow an estimation of sPAP in connection with SBP [47,52]. Likewise, right-to-left shunting and low velocities across the shunt are indicative of a supra-systemic sPAP [50].

### 1.2. Estimation of mPAP and Diastolic PAP (dPAP)

If pulmonary regurgitation (PR) can be interrogated with CW-Doppler in the parasternal short axis (PSAX) view, mPAP and dPAP can be estimated from the maximum (early-diastolic) and minimum (end-diastolic) PR velocity, respectively, using the simplified Bernoulli equation taking into account mean RAP [53]. The use of the PR gradient may be preferred for the estimation of mPAP and dPAP, especially in cases when TRV is unreliable [53]. However, the reliability of PR velocities may be affected by significant PS or severe PR [47]. In patients with, e.g., unrepaired pulmonary or tricuspid atresia, Doppler gradients across valves are not applicable and pulmonary blood supply commonly occurs through a PDA or major aortopulmonary collateral arteries (MAPCAs) [54]. Large MAPCAs can lead to excessively increased blood flow to a specific lung segment, resulting in segmental PH and PVD over time [47,55]. Although echocardiography is not sufficient for firmly establishing the diagnosis of segmental PH, CW-Doppler can be used to interrogate flow through collateral vessels or surgical shunts (if present) for estimating gradients between the aorta and the pulmonary vessels [55]. In patients where the sole source of blood supply of the lungs is a (modified) BT shunt, the diastolic BT flow velocity was shown to reliably predict mPAP in patients with complex CHD [56]. Furthermore, segmental PA dilation or the presence of large-caliber collateral vessels to the lung may indicate segmental PH [55].

### 1.3. PA Acceleration Time (PAAT)

Another useful parameter for the estimation of PAP and PVR is the PAAT, defined as the interval in milliseconds from the onset of ejection to peak flow velocity. For obtaining the PAAT, the PA forward flow velocity profile, obtained in the RVOT with pulsed wave (PW) Doppler just proximal to the pulmonary valve, can be used (Figure 1A) [37,57]. The normal PW Doppler profile in the PA is smooth and parabolic, whereas “notching” of the PW Doppler envelope was shown to be associated with increased PVR in adults with PAH [58,59]. A linear inverse relationship between PAAT and mPAP was shown in adult patients with a PAAT < 100 ms [60]. Likewise, in children with PAH-CHD, a negative correlation between PAAT and catheter-derived parameters as sPAP, mPAP, dPAP and PVRi has been established [61]. As PAAT varies with heart rate, pediatric reference values should be used in children [46,62].

Besides PAAT alone, the ratio of PAAT and RV ejection time (RVET) has been shown to be reduced in adults and children with PH and to highly correlate with mPAP [63,64]. Especially important for the assessment in the pediatric population, the PAAT/RVET ratio is less dependent on age, body surface area and heart rate [63,65,66].

### 1.4. RVOT Velocity Time Integral (VTI)

For estimation of PVR, the RVOT VTI can be obtained from PSAX by placing a PW Doppler sample in the RVOT just proximal to the pulmonary valve [67]. Age-related pediatric RVOT VTI values are available [67]. An association between an increased PVR and a decreased RVOT VTI has been shown in adults and children with PH [68,69]. Moreover, a correlation between the ratio of TRV/RVOT VTI and invasively measured PVR has been demonstrated in adults and children [70,71,72]. Specifically, a TRV/RVOT VTI ratio of 0.14 has been shown to provide a sensitivity of 97% and a specificity of 93% for PVR values of > 6 WU, whereas TRV/RVOT VTI ratios of 0.17 provided a sensitivity of 79% and a specificity of 95% for PVR values of > 8 WU in children with CHD and left-to-right shunts independent of age or RVOT diameter [67,73].

### 1.5. Systolic-to-Diastolic Duration Ratio

The systolic (S) to diastolic (D) duration ratio reflects the global RV function. By using TR duration from Doppler flow, durations of S and D are measured in the apical four-chamber view (Figure 1B). The duration of the TR flow reflects S, whereas D duration is considered from termination to onset of TR [37]. As an independent geometric index, it provides a simple and widely available echocardiographic method even in children and adults with biventricular CHD [74,75]. In patients with severe PH, RV contraction is significantly prolonged (and consequently diastole is shortened) despite a shorter RV ejection time leading to an increased Doppler-derived S/D duration ratio compared to healthy controls [76]. Of note, the duration of S and D is heart rate dependent, and thus, worsened S/D duration ratios are especially seen in children with PH during tachycardia [37,77]. An S/D duration ratio > 1.4 has been shown to inversely correlate with survival in pediatric PH [77].

In children with HLHS, S/D duration ratios after the Norwood procedure (first stage of palliation) were significantly higher compared to those after Glenn- or Fontan-Procedure (second and third stage of palliation; after the latter, a Fontan circulation is present). However, S/D durations after Glenn-/Fontan-Procedure were demonstrated to be significantly higher as compared to healthy controls [75]. Moreover, an AVV S/D duration ratio > 1.1 is a strong predictor of a ventricular end-diastolic pressure (VEDP) ≥ 10 and was associated with poor prognosis in adult patients with Fontan circulation [78,79].

### 1.6. Multiparametric Approaches Including Apical TAPSE and Subcostal TAPSE

TAPSE, usually measured in M-Mode in the apical four-chamber view, assesses the longitudinal excursion of the tricuspid annulus towards the apex during systole (Figure 2A) [80]. Being vastly independent of heart rate, TAPSE is an accepted parameter for the evaluation of RV systolic function in children and adults with PH, with and without CHD [81,82,83]. Reduced TAPSE values have been shown in children after surgical repair of, e.g., VSD and TOF, as well as in children with HLHS after Fontan palliation [82,84,85,86]. Of note, TAPSE is a preload-dependent parameter and should, therefore, be interpreted with caution, as significant TR volumes may lead to false higher TAPSE values due to larger systolic and diastolic RV volumes [86].

Contemporary echocardiographic assessment of RV function in patients with PH requires a multiparametric approach that includes TAPSE to avoid overreliance on a single parameter. For example, TAPSE/sPAP ratio values have been shown to inversely correlate to New York Heart Association functional class (NYHA FC) and thus might be helpful in predicting outcomes in children and adults [82,83,84,85,86,87,88,89]. Furthermore, the TAPSE/sPAP ratio was shown to be affected by RV diastolic stiffness in adults with severe PH [90] and reduced TAPSE/sPAP ratios correlated with RV restrictive physiology in children after surgical repair of TOF [91].

In critically ill patients following cardiac surgery, the assessment of TAPSE in the apical four-chamber view is often hampered by extensive wound dressings [92]. Reference values and z-scores for subcostal TAPSE (S-TAPSE) were recently provided for adult and pediatric patients who may assist in identifying patients with reduced RV function [92,93]. In a small cohort of children with PH-CHD, reduced S-TAPSE values indicated impaired systolic RV function [92].

Importantly, it should be noted that TAPSE is preserved for a long time in the disease process, especially in younger children, despite very high PAP, and as such, TAPSE is a typical echocardiographic variable that when used as a single surrogate, can lead to a gross misinterpretation of systolic function.

### 1.7. RV Fractional Area Change (FAC)

The RV FAC, recommended by recent guidelines for the evaluation of RV systolic function [1,2,3], reflects longitudinal and radial RV function and is calculated as (diastolic RV area-systolic RV area)/diastolic RV area [50]. A correlation between RV FAC and RV ejection fraction (RVEF) and TAPSE has been shown in adults with IPAH [94]. In pediatric PAH, an RV FAC < 25% may indicate clinical worsening [95,96,97]. Moreover, a decreased RV FAC was associated with the risk of death in adult and pediatric IPAH patients [93,94].

FAC use for assessing RV systolic function is limited by several factors: FAC is dependent on preload and has been shown to be less reproducible than TAPSE [98,99,100]. Furthermore, incomplete delineation of the RV is common, especially in the presence of RV dilation, and RV FAC analysis is therefore associated with higher inter- and intraobserver variabilities [100]. Moreover, even though pediatric reference values for RV FAC in the healthy population exist, they do not reflect the situation in patients after surgical repair of CHD [101,102].

### 1.8. RV Tissue Doppler Imaging

Tissue Doppler imaging (TDI) allows for the assessment of systolic and diastolic ventricular function by measuring myocardial velocities by PW-Doppler in the apical four-chamber view [103,104]. The PW-Doppler sample should be placed at the level of RV lateral tricuspid annulus, the basal IVS and the LV lateral mitral annulus [50]. The systolic longitudinal function of the RV and LV is reflected by the myocardial systolic wave (S’), whereas diastolic ventricular function is denoted by the early diastolic wave (E’) and the late-diastolic wave (A’), with the latter reflecting atrial contraction [37,50]. PW-Doppler TDI determines peak myocardial velocities, whereas color TDI assesses mean velocities, which are lower compared to PW TDI values (Figure 2B) [50]. As tissue Doppler velocities vary with age and heart rate, age-related reference values must be applied for pediatric patients [105].

Especially beneficial in patients with operated PH-CHD, TDI is independent of chamber geometry, which permits its use in any chamber configuration [106]. Reduced tricuspid S’ values have been shown to be related to invasively measured mPAP and PVR [107] and to correlate with RV FAC and TAPSE in adult patients with PH [108]. A good correlation between tricuspid S’ and the RVEF has been demonstrated in an adult PH population that included patients with CHD [109]. Likewise, in children, RV S’ values measured at the tricuspid valve have been demonstrated to be substantially impaired in pediatric PH-CHD, with S’ values continually decreasing with longer duration of PH [110].

In adult patients with RV pressure overload due to severe PH, RV tricuspid S’ was demonstrated to be lower as compared to patients with RV volume overload due to, e.g., an ASD (only left-to-right shunting, no patients with right-to-left shunting), in which elevated tricuspid S’ and lower tricuspid E’ values were shown [111,112]. Accordingly, in children with an ASD and RV dilatation, elevated tricuspid S’ values have been reported before percutaneous closure of the defect, a phenomenon that normalized within 24 h after closure [111]. Furthermore, in these patients, tricuspid E’ and E’/A’ values negatively correlated with sPAP, mPAP and dPAP, respectively [113].

### 1.9. RV Strain Measurements

Strain measurements provide substantial information on RV and LV global and regional function. The technique is angle independent and less load-dependent than classic parameters of ventricular function. [114,115]. This is particularly of interest in CHD as wall motion abnormalities and very heterogeneous loading conditions are a common finding in affected patients [116]. The strain is a dimensionless measure of myocardial deformation, whereas the strain rate is defined as the rate of deformation over time [37]. In children and adults, specific reference values should be used to call a measurement, either abnormal or normal [117,118].

In adults with PAH, RV global longitudinal peak systolic strain and strain rate have been shown to be lower with varying degrees of PAH than controls [119]. Likewise, in children with IPAH, a decrease in RV longitudinal deformation, transverse shorting, and post-systolic shortening has been shown [120]. Moreover, RV longitudinal free-wall strain has been shown to be prognostic of adverse clinical outcomes in children as well as in adults with PH [121,122]. In adult patients with PAH-CHD, significantly lower strain, and strain rates of the lateral tricuspid annulus have been shown in comparison to controls [123]. Furthermore, reduced regional peak systolic strain and strain rates have been demonstrated in the apical segments of the RV free wall in patients with a systemic RV after the Senning procedure [124]. In patients after TOF repair, systolic strain and strain rate values in the basal, mid and apical segments of the RV free wall and the IVS have been shown to correlate with the degree of PR [125]. However, it must be taken into account that the application of normative 2-D strain cutoff values is hampered by multiple factors, e.g., regurgitant volumes or RV dilation may cause increased or decreased 2-D strain values, respectively [126]. While current recommendations define RV strain as the mean value of the RV free wall only [3,37], it is often useful to also include the septal regions into the assessment of RV-LV interaction [24,127]. Besides 2-D strain variables, relevant correlations between RV failure and 3-D strain parameters were demonstrated in children and in adults [43,128]. However, the clinical application of 3-D strain measurements in children with PH, especially in those with CHD, requires further research.

### 1.10. RV Diastolic Function

In both adults and children with PH, RV diastolic dysfunction and elevated RAP are associated with increased mortality [18,99,129]. For echocardiographic evaluation of RV diastolic function, tricuspid inflow velocities (E, A, E/A), TDI at the tricuspid valve (E’, A’, E’/A’), deceleration time and the isovolumic relaxation time (IVRT) should be obtained in the apical four-chamber view using PW Doppler. Moreover, the ratio of tricuspid E/E’ can be used for assessing potential RV diastolic dysfunction [118]. However, tricuspid inflow velocities E and A may be confounded in the presence of severe TR [50], and all of these variables are preload dependent.

In children with PH, lower E’ velocities have been shown as compared to controls [130]. The tricuspid valve E’ velocity correlated significantly with invasively measured mPAP and RV end-diastolic pressure (RVEDP) in children with IPAH [131]. In accordance, in a cohort of children with PAH-CHD after surgical repair of intracardiac shunts, tricuspid E’ and S/D duration ratio correlated significantly with invasively measured RVEDP [26]. However, different from the LV, in which the mitral E/E’ ratio is typically used to evaluate LV end-diastolic pressure (LVEDP), the tricuspid E/E’ ratio did not correlate with RVEDP in children with existing intracardiac shunts [26]. As the E’ velocity is load-dependent, this may be due to altered RV preload by left-to-right shunting [26,132].

RA strain measurements are also useful indicators of RV diastolic function. In adults and in children with PH, all phases of atrial function (reservoir-, conduit- and pump-phase) have been shown to be impaired, which likely reflects RV failure and overload [20,133]. Worse reservoir function has been shown to be associated with worse clinical outcomes in adult and pediatric PH [20,134]. However, in children, RA pump function is more preserved at the time of diagnosis of PH as compared to adults, and an impaired RA pump function has been shown to predict adverse clinical events in children with PH [20]. Moreover, adults with PH who lost RA function due to atrial fibrillation have been shown to deteriorate clinically with poor outcomes [135]. The development of arrhythmias is also a well-known problem in patients with CHD (e.g., TOF), as repeated cardiac surgical procedures and atrial enlargement contribute to atrial arrhythmias [136]. Importantly, RA function cannot be assessed echocardiographically in the presence of significant ASDs or TR.

### 1.11. RV/LV Interaction

VVI refers to the impact of alterations in function, filling, pressure and volume related changes in geometry and synchrony that one ventricle shows in response to pathologic changes that primarily affect the other ventricle [24,25,137]. The RV/LV ratio, measured in the PSAX at end-systole at the LV papillary muscle level, reflects the extent of LV compression imposed by a hypertensive RV (Figure 3). In pediatric and in adult PH patients, significantly higher RV/LV ratios were demonstrated when compared to control subjects [138,139]. An RV/LV ratio >1 was associated with adverse clinical events [139,140]. Of course, the assessment of the RV/LV ratio assumes biventricular circulation and is therefore not applicable in patients with single ventricles.

The end-systolic LV eccentricity index (LVesEI), measured in the PSAX, is defined as the ratio of the minor axis of the LV parallel to the septum divided by the minor axis perpendicular to the septum and is used to evaluate the LV compression for hemodynamic assessment in PH patients (Figure 3) [37,141]. In children, LVesEI has been shown to be higher in patients with IPAH as compared to those with PH-CHD [99]. Pediatric PH-CHD patients with an increased LVesEI were found to have significantly smaller LA areas and an increased sPAP/SASP (pulmonary to systolic systemic arterial pressure) ratio, and an increased PVRi [138]. An LV end-diastolic EI > 1.7 has been shown to have prognostic value in adults with PH and to correlate well with invasive measurements of PAP [142].

However, the maximal leftward septal shift in PH often occurs at the time of mitral valve opening (early LV diastole) [24]. In accordance, a short mitral inflow time and a reversed mitral inflow E/A ratio are frequently present in a pattern of delayed relaxation [24,143,144]. Referring to this observation, a new TTE measure, entitled the post-systolic maximal eccentricity index (LVpsEIM), which is calculated at maximal septal flattening, was recently introduced for children [25,143,144]. LVpsEIM measures were significantly increased in children with PH and correlated well with invasive hemodynamics and outcome measures [25].

By using strain measurements, a negative impact of RV dilation on LV circumferential deformation, but not on longitudinal or radial deformation, has been demonstrated [145]. Furthermore, LV torsion seems to be impaired predominantly due to reduced basal rotation in states associated with RV volume load [80,146]. In young adults, acute unloading of the RV after interventional closure of ASDs improved LV twist by increasing basal rotation [146]. An association between LV myocardial mechanics and invasive hemodynamics, and RV myocardial function has been demonstrated in children with PH [147]. Furthermore, a correlation between LVesEI and basal septum strain has been reported [147].

However, the RV/LV ratio, the LVesEI as well as the LVpsEIM should be interpreted cautiously, especially in patients with PH/PAH after surgical repair of CHD as the structure and shape of the IVS may be affected by the procedure (e.g., due to ventricular patches) [148]. Furthermore, other causes of RV pressure overload, which could potentially lead to IVS flattening, must be excluded (e.g., RVOT obstruction, pulmonary stenosis or severe PR) [47,148] to avoid false-positive results.

## 2. Echocardiographic Features of the LV in PH-CHD

### 2.1. LV Systolic Function

LV systolic function can be estimated by LVEF determination using the biplane Simpson formula. In pediatric patients with severe PH, a decreased LVEF has been shown to be associated with an increasing ratio of sPAP/sSAP and an increasing PVRi [138]. Although LVEF is one of the most widely used parameters to assess LV systolic function, it is limited in its ability to determine ventricular contractility by load dependency [149]. Moreover, irregular LV geometry, which is common in CHD, especially after surgical repair or in severe PH due to RV pressure overload and LV compression, leads to inaccuracies in calculating LVEF [150]. Strain measurements may reveal LV systolic dysfunction even in patients with normal LVEF. Decreased LV septal longitudinal strain has been shown in adult PAH patients compared to controls [151]. In adult PAH-CHD patients, lower strain and strain rates of the LV free wall and the septum have been demonstrated most likely due to ventricular interdependence in biventricular circulations [123]. In children with severe PH, despite reduced basal septal LV longitudinal strain, increased free-wall longitudinal strain maintained global longitudinal strain within normal limits [147].

Severe septal shift may lead to obstruction of the LV-mid cavity or LVOT and therefore to decreased CO. CO can be estimated by the following equation: CO = (LVOT diameter/2)^2^ × 3.14 × VTI (LVOT) × HR [50]. The LVOT diameter should be determined from the parasternal long-axis view and the LVOT VTI value by using PW Doppler measurement from the apical four-chamber view. Reference intervals for obtaining LVOT VTI in (healthy) children are available [152].

### 2.2. LV Diastolic Function

The differentiation between normal and abnormal LV diastolic function is challenging as diastolic measurements are heavily influenced by age, heart rate and preload, among other factors [116]. LV diastolic function can be examined by using PW Doppler measurements of the mitral inflow and the flow of the pulmonary veins combined with TDI measurements of the lateral and medial mitral annulus and the estimation of LA pressure [132,153].

The mitral Doppler inflow is assessed in the apical four-chamber view with the curser placed across the mitral valve just inside the LV [153]. Early LV diastolic filling is represented by the mitral E wave, whereas the later mitral A wave demonstrates LA contraction [116,154]. Generally, a reduced mitral E/A ratio < 0.9 may indicate an impaired LV relaxation, while an increased mitral E/A ratio > 2 may indicate a reduced LV compliance in adults [132]. In children, mitral E wave velocities increase within the first year of life, whereas mitral A wave velocities decrease during infancy and childhood. As a result, early to atrial phase parameters like mitral E/A ratio and the LA filling fraction undergo significant changes throughout maturation. [155]. Therefore, reference values for age-related changes in diastolic LV flow parameters should be used in children [155,156]. Of note, in healthy children and adults, diastolic parameters of LV filling undergo age-related changes due to a slowing of LV relaxation [132]. However, there is significant inter- and interindividual heterogeneity in mitral inflow velocities, and evaluation of LV diastolic indices should thus be performed cautiously, especially in younger children [156]. The pulmonary venous flow should be assessed by placing the PW Doppler in the right or left upper pulmonary vein [132]. Thereby, pulmonary venous S, D and peak atrial reversal flow velocities (AR) can be obtained. Increased AR velocities may indicate an increased LA pressure [116].

By TDI measurements, mitral S’, E’, and A’ velocities can be determined; however, in pediatric patients, normal values and z-scores should be used [116,157]. The E/E’ ratio is, compared to the E/A ratio, less preload-dependent and was shown to strongly correlate with invasively measured PA wedge pressure (PAWP) values, which approximates mean LA pressure in the absence of pulmonary vein stenosis [154]. The American Society of Echocardiography (ASE) recommends a mitral E/E’ ratio of >14 as the cutoff value for diastolic dysfunction in adults [132]. However, the use of mitral E/E’ values to assess LV filling pressures is maybe limited in pediatric patients as approximately 50% of children with cardiomyopathy have been shown to exhibit E/E’ values within age-adapted reference ranges [26,156].

For LA pressure estimation, the assessment of the LA maximum volume index from the apical four-chamber view at end-systole should be preferred over measurements of linear dimensions [116,132]. A LA volume index > 34 mL/m^2^ may indicate LV diastolic dysfunction in adults [132]. Age-related normal values of the LA volumes should be used in children [158]. An increased LA volume has been shown to correlate with worse hemodynamics and poor outcomes in children with CHD, such as VSD [159]. Bowing of the interatrial septum rightwards is also suggestive of high LA pressure [160]. In children with precapillary PH under generalized anesthesia, an increased pulmonary vein AR wave duration, reduced mitral E waves and an increased mitral E/E’ ratio may indicate a decreased LV compliance [143,144]. Impaired LV diastolic function was furthermore related to invasive hemodynamics, leftward septal shift and prolonged RV systole [143,144].

After percutaneous ASD device closure, mitral E wave and mitral E/E’ ratio values significantly increased, which indicated elevated LV filling pressures in a small cohort of adult patients [161]. In contrast, velocities did not change after percutaneous ASD device closure in pediatric patients, which suggests a preserved diastolic function in children providing another rationale for earlier defect repair [162].

Echocardiographic evidence of systolic and diastolic ventricular dysfunction has also been demonstrated by TDI measurements in patients with Fontan physiology, with TDI velocities being lower in both left and right systemic ventricles as compared to normal age-related values. However, morphologic RVs show more severely depressed TDI values than LVs in patients with single-ventricle physiologies [163]. In children with morphologic right single-ventricles, higher PW Doppler-derived pulmonary vein S’ velocities have been shown after initial staged palliation (Norwood procedure), in comparison to healthy infants, in which diastolic venous emptying is predominantly seen by 1 year of age [164,165].

In addition, a higher LV free wall E/E’ ratio (≥12) is associated with a reduced peak oxygen uptake (VO2peak) in adult patients after Fontan palliation [166].

## 3. Exercise Echocardiography

Commonly, echocardiography is performed at rest and can thus only partially aid in understanding the pathophysiology of the often debilitating exercise intolerance experienced by patients suffering from PAH and RV dysfunction [32]. While exercise testing such as 6 min walk test is predictive of outcome in children with PH, its results do not correlate to echocardiographic assessment parameters at rest, hampering the use of resting echocardiography in estimating cardiac function during activity and exercise [167]. Exercise echocardiography in the assessment of LV ad RV function has recently been used successfully to describe myocardial exercise reserve in healthy children [168,169,170], pediatric athletes [171] but also children with CHD [172,173].

The assessment of RV function during exercise stress (ventricular reserve) has been used in adult PAH [174] to explain the lack of correlation between resting RV dilatation, function and NYHA class and exercise capacity [34]. High RVSP, although often used in this context, was described to be not predictive of reduced exercise capacity and RV function. RVSP can rise in healthy adults during exercise to values of 50 mm Hg [175], and the ability to increase RVSP during exercise in patients with PAH was described to represent a positive contractile response of the RV [34].

A pathophysiological study questioned the hypothesis that PAP changes indeed indicated RV contractility changes [176] and concluded that rest to exercise responses in PAP only depict this response in heart rate changes. Others described that an increase of cardiac output during dobutamine exercise in adult PAH patients was associated with an improvement of the TAPSE, an increase in heart rate, but with a decrease of TRV [177]. Echocardiographic indices of RV contractility, such as the TAPSE/sPAP ratio was found not to be associated with an increase of RV function [177], suggesting an absence of contractile reserve of adult PAH patients during cardiopulmonary exercise testing [176,177].

Parameters such as PW TDI and 2-D strain can describe myocardial exercise reserve, and normal reference values for children exist [168,169,170]. Importantly, there is evidence for a pathologically cardiac exercise response in children with CHD, such as an altered frequency relationship during exercise in children with tetralogy of Fallot [173]. While exercise echocardiography studies have not been published yet in children with PH, less load-dependent parameters such as PW TDI or 2-D strain have the potential to assess cardiac function also in the pediatric PH population.

## 4. Conclusions

In PH-CHD, differentiation between an elevated PAP with or without PVD and appropriate etiologic classification of disease entities represent critical prerequisites for optimal management. The new WSPH classification (2018) may assign a patient to more than one PH group (e.g., HPAH and interstitial lung disease or PAH-CHD with a significant postcapillary component). Screening for the presence of PH during routine TTE should be performed in every CHD patient as PH/PAH may develop even after surgical repair, and this review highlights the most useful and predictive echocardiographic parameters and their specific virtue. This review also describes that many echocardiographic variables and indices which are recommended for screening of (suspected) PH in international guidelines do not apply to patients with CHD, and the rationale for specific echocardiographic variables to monitor PH in CHD are described. In addition, as exercise capacity represents a major potential outcome variable in children with PH, this review provides an overview of the use of exercise stress echocardiography as a novel tool to monitor myocardial exercise performance.

## Figures and Tables

**Figure 1 diagnostics-11-00049-f001:**
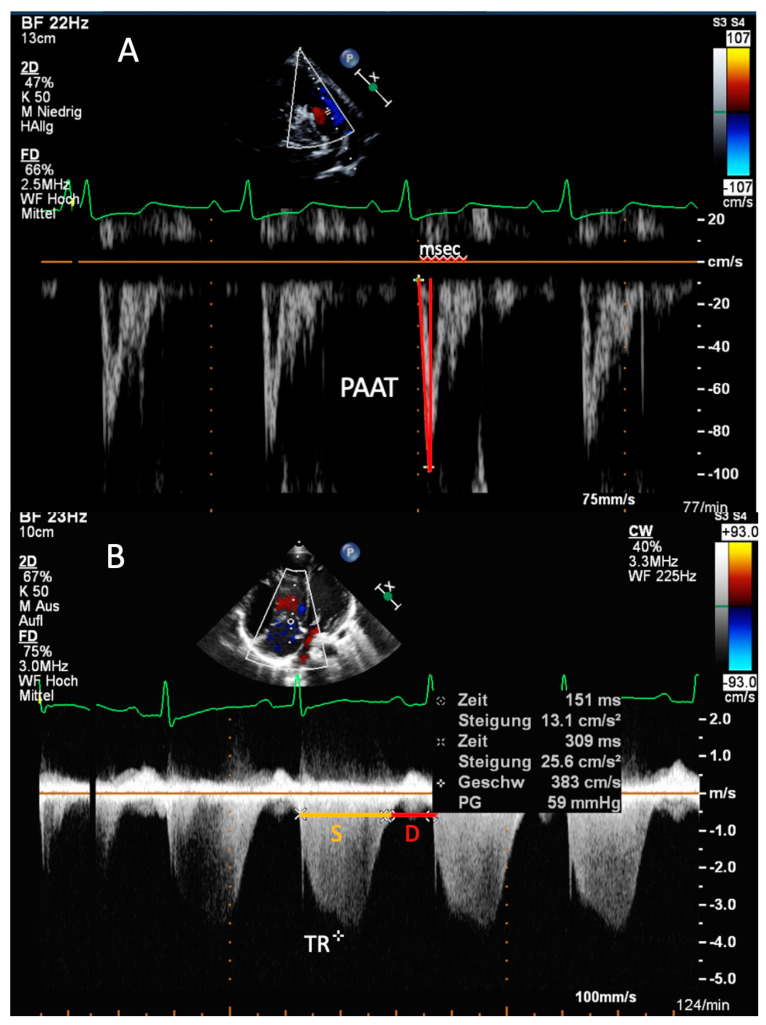
Transthoracic imaging. (**A**) Parasternal short-axis view. Measurement of the pulmonary artery acceleration time (PAAT) in an 8-year-old male patient with pulmonary arterial hypertension associated with congenital heart disease (PAH-CHD). The red lines mark the rapid acceleration to peak flow velocity in early systole, followed by a fast deceleration in mid-systole. Compared to healthy subjects, the PAAT of 55 ms measured over 4 circles is reduced in this patient. (**B**) Apical four-chamber view. Determination of Doppler-derived systolic (S) to diastolic (D) duration ratio in a 12-year-old patient with idiopathic pulmonary arterial hypertension by using tricuspid regurgitation (TR). The yellow line marks the S duration, and the red line the D duration of the cardiac circle. In this patient, the S/D ratio is increased (S/D ratio = 1.9).

**Figure 2 diagnostics-11-00049-f002:**
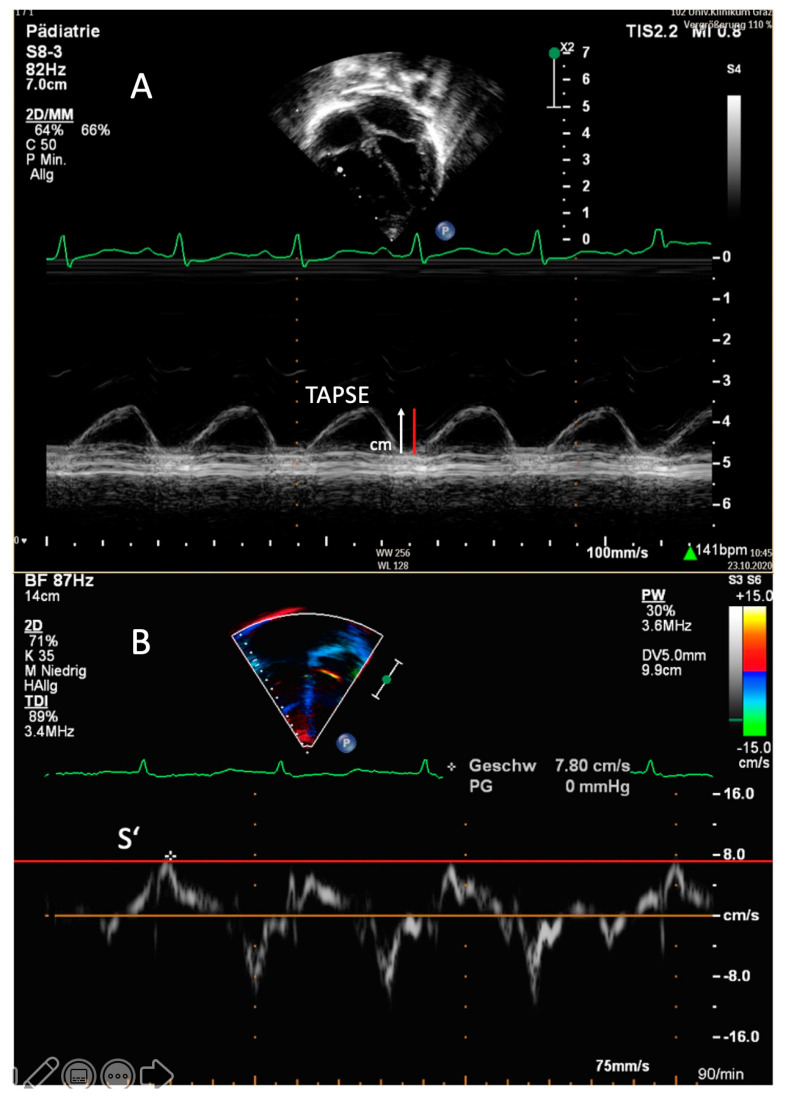
Transthoracic imaging. (**A**) Apical four-chamber view. Determination of the tricuspid annular plane systolic excursion (TAPSE) in a 1-year-old patient with idiopathic pulmonary arterial hypertension in M-mode. The red line demonstrates a TAPSE value that is abnormally low in this patient (1.1 cm; z-score −3). (**B**) Apical four-chamber view. Right, ventricular tissue Doppler imaging (TDI) with the PW curser placed at the lateral tricuspid annulus in an 8-year-old patient with pulmonary arterial hypertension associated with congenital heart disease (PAH-CHD). In this patient, the values of the tricuspid peak systolic excursion (S’) are reduced, depicted as the red line.

**Figure 3 diagnostics-11-00049-f003:**
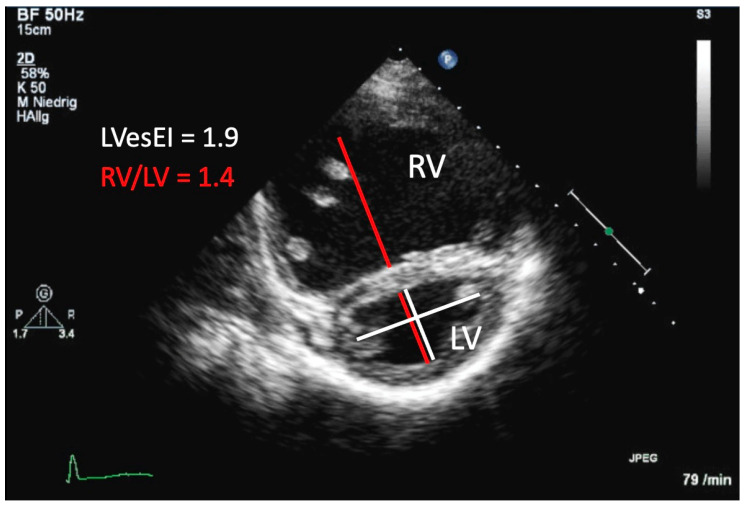
Transthoracic imaging. Parasternal short-axis view of the right ventricle (RV) and the left ventricle (LV) in a 6-year old girl with idiopathic pulmonary arterial hypertension. The LV appears D-shaped due to flattening of the interventricular septum caused by increased pressure of the RV. The RV/LV ratio was derived at end-systole (red lines). For obtaining the LV eccentricity index (LVesEI), the shorter white line represents the septal-lateral dimension of the LV, whereas the long white line represents the anterior–posterior dimension.

**Table 1 diagnostics-11-00049-t001:** Hemodynamic definitions of pulmonary arterial hypertension/pulmonary hypertension; (* Calculate for mean TPG mPAP—mLAP or PAWP).

Hemodynamic PH Definitions (According to the Recent WSPH, Nice 2018)	
**Pre-capillary PH/PAH** (Clinical groups 1, 3, 4, 5)	mPAP > 20 mmHg (in children > 3 months of age at sea level) PAWP or LVEDP ≤ 15 mmHg PVRi ≥ 3WU x m^2^ (may be < 3 WU m^2^ in CHD with left-to-right shunting)
**Isolated post-capillary PH** (Clinical groups 2, 5)	mPAP > 20 mmHg (in children > 3 months of age at sea level) PAWP or LVEDP > 15 mmHg PVRi < 3WU x m^2^
**Combined pre- and postcapillary PH** (Clinical groups 2, 5)	mPAP > 20 mmHg (in children > 3 months of age at sea level) PAWP > 15 mmHg PVRi ≥ 3WU x m^2^
**PVD for circulations with TCPC** (e.g., Fontan physiology)	Mean TPG^*^ > 6 mmHg or PVRi ≥ 3WU x m^2^

**Abbreviations:** CHD, congenital heart disease; mLAP, mean left atrial pressure; mPAP, mean pulmonary artery pressure; PAH, pulmonary arterial hypertension; PH, pulmonary hypertension; PVD, pulmonary vascular disease; PVRi, pulmonary vascular resistance index; PAWP, pulmonary arterial wedge pressure; TCPC, total cavopulmonary connection; TPG, transpulmonary pressure gradient; WSPH, World Symposium on Pulmonary Hypertension.

**Table 2 diagnostics-11-00049-t002:** Clinical classification of pulmonary arterial hypertension/pulmonary hypertension associated with congenital heart disease (modified from Simmoneau et al. [1]).

PH Group	Subgroup
**Group 1**	**1.4.4 PAH-CHD** (A) Eisenmenger syndrome (B) PAH associated with systemic-to-pulmonary Shunt (C) PAH and coincidental/small defects (D) PAH following surgical repair of CHD/defect closure
**Group 2**	**2.4 Congenital post-capillary obstruction lesions** (e.g., pulmonary vein stenosis, mitral/aortic stenosis, coarctation of the aorta)
**Group 3**	**3.5 Developmental lung disorders** (e.g., bronchopulmonary dysplasia, congenital diaphragmatic hernia)
**Group 4**	**4 PH due to pulmonary artery obstructions** (congenitally or acquired after surgical repair of CHD)
**Group 5**	**5.4 Complex CHD** (e.g., segmental PH, single ventricle/TCPC, scimitar syndrome)

**Abbreviations:** CHD, congenital heart disease; PAH, pulmonary arterial hypertension; PH, pulmonary hypertension; TCPC, total cavopulmonary connection.

**Table 3 diagnostics-11-00049-t003:** Advantages, disadvantages and considerations of echocardiographic measurements in patients with PH/PAH-CHD.

Parameter	Advantages	Disadvantages/Considerations in PH-CHD
**Peak TRV**	Estimation of sPAP taking into account mean RAP, easily to perform; Is of significant prognostic value for PH	Assumes the absence of RVOTO or PS; AVV regurgitation does not reflect sPAP in patients with TCPC; Dependent on RV systolic function
**Early PRV**	Estimation of mPAP taking into account mean RAP; Should be preferred when TRV is unreliable	PR required for PRV measurements; May be under-/overestimated in severe PR
**Late PRV**	Estimation of dPAP taking into account mean RAP; Independent of RV systolic function	
**PAAT**	Can be measured in most of PH patients; PAAT < 100 ms in adults—PH is likely; Existing reference values in children	Possible pulmonary valve artifacts; Heart rate dependent (PAAT/RVET ratio is less dependent on age, BSA and heart rate)
**TRV/ RVOT VTI**	Allows noninvasive assessment of PVR in adults and children	Calculation of this ratio assumes a circular geometry of the RVOT
**S/D ratio**	S/D duration ratio > 1.4—PH is likely; Geometric independent ratio	Requires presence of defined TR onset/end; Heart rate dependent parameter
**TAPSE/ sPAP ratio**	Applicable in children and adults	Does not take into account segmental or radial RV function and contractility
**S-TAPSE**	Useful to determine in critically ill patients; Existing adult and pediatric reference values;	Useful when 4CV is not possible due to e.g., wound dressing in the ICU setting
**FAC**	Reflects longitudinal and radial RV function	Load dependent; may not apply in patients with CHD especially after surgical repair
**RA function**	RA strain values reflect RV diastolic function; Atrial arrhythmias are associated with poor outcome	Cannot be used in patients with ASDs or severe TR
**RV/LV TDI**	Reflects RV/LV systolic and diastolic function and VVI; Independent of chamber geometry; Existing reference values in children; Decreased in adult/pediatric PH-CHD patients	Variability with different loading conditions; Assessment of motion in a single dimension;
**RV/LV ratio**	Reflects extent of LV compression; Ratio > 1—PH is likely;	Cannot be used in PH patients with significant left-to-right shunt lesions; Other causes of IVS flattening must be excluded (e.g., RVOTO, PS, PR) Structure and shape of IVS may be altered postoperative (e.g., VSD Patch); Not applicable in patients with single ventricle
**LVEI**	Reflects extent of LV compression; Index > 1.7 indicates adverse outcome in adult PH	
**LVpsEIM**	Reflects maximal extent of LV compression; Significantly increased in pediatric PH	
**LVEF**	Reflects LV systolic function; Correlates with invasively measured parameters e.g., PVRi, sPAP/sSAP ratio	Load dependent; assumes regular LV geometry; Not applicable in cases of LV compression
**RV/LV Strain**	Reflects global and regional RV/LV function; Existing reference values in children; Decreased in adult/pediatric PH-CHD patients	Interpretation of strain data is often complicated especially in patients with CHD
**LV diastolic function**	E/e’ ratio > 14 indicates elevated LAP; LA volume index > 34 mL/m^2^ indicates LV diastolic dysfunction in adult patients; Bowing of LA septum indicates high LAP	Dependent on age, heart rate and preload; Often not reliable in children

**Abbreviations:** ASD, atrial septal defect; AVV, atrioventricular valve; BSA, body surface area; CHD, congenital heart disease; dPAP, diastolic PAP, FAC, fractional area change; mPAP, mean PAP; LA, left atrial, LAP, LA pressure; LV, left ventricle; LVEI, LV eccentricity index; LVEF, LV ejection fraction; LVpsEIM, LV post-systolic maximal eccentricity index; PAAT, pulmonary acceleration time; PAP, pulmonary artery pressure; PH, pulmonary hypertension; PR, pulmonary regurgitation; PRV, PR velocity; PS, pulmonary stenosis; RA, right atrium; RV, right ventricle; RVET, RV ejection time; RVOT, RV outflow tract; RVOTO, RVOT obstructions; S/D, systolic/diastolic duration ratio; sPAP, systolic PAP; RAP, right atrial pressure; S-TAPSE, subcoastal tricuspid annular peak systolic excursion; TCPC, total cavopulmonary connection; TDI, tissue doppler imaging; TR, tricuspid regurgitation; TRV, TR velocity; VTI, velocity time integral; VVI, ventricular-ventricular interactions; 4CV, 4-chamber view.

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
