# Peer review of "Echocardiography for the Assessment of Pulmonary Hypertension and Congenital Heart Disease in the Young"

_diagnostics, 2020, doi:10.3390/diagnostics11010049_

Round 1
Reviewer 1 Report
The authors presented an interesting review on the echocardiographic assessment of pulmonary circulation and right ventricular function. The vast majority of the article refers to data from the pediatric population, and I think it would be better to modify the title to include this information.
The review article covers the issues described comprehensively, contains a lot of current literature, and is an excellent material for people dealing with echocardiography or those who want to be interested in assessing the right ventricle in children (and adults) with congenital heart disease. Some minor remarks should be made:
1.The findings of the 6th WSPH actually introduced PVR, not PVRI, into the definition of PH. These values ​​and cut-off points (3 Wood Units) should not be used interchangeably.
2. Sentence "Thus, PH or PAH with only slightly elevated PVR index is neither synonymous nor indicative of PVD in patients with CHD, e.g., systemic-to-pulmonary shunts lead to an enhanced pulmonary blood flow and consequently to an increased mPAP with or without PVD, as PVR (in children: PVR indexed [PVRi]) may be within the normal range "is not entirely clear. As far as the authors are concerned with pulmonary hypertension with the hyperkinetic flow, they should not use the term PAH, but simply PH.
3. In Table 3, among the limitations of the "peak TRV" assessment, I would add the necessity to estimate the right atrial pressure for the SPAP calculation and the TRV value's dependence on the right ventricular systolic function.
4. The relationship between the pulmonary ejection acceleration time and the pulmonary artery pressure is modified by the type and location of the primary point of resistance in the pulmonary circulation. It is mentioned in the paper by A Torbicki et al. (Proximal pulmonary emboli modify the right ventricular ejection pattern. Eur Respir J. 1999 Mar; 13 (3): 616-21) and worth mentioning
5. The etiology of the D-shape sign in patients with pulmonary hypertension is not entirely clear. Authors write that the left ventricle is compressed. At the same time, the invasively measured left ventricular diastolic pressure in patients with severe pulmonary hypertension is normal or even low. This indicates that the left ventricle is rather underfilled and collapsed.
Author Response
Response to Reviewer #1
The authors presented an interesting review on the echocardiographic assessment of pulmonary circulation and right ventricular function. The vast majority of the article refers to data from the pediatric population, and I think it would be better to modify the title to include this information.
RESPONSE: We agree with the reviewer that our data refers predominantly to the pediatric population and included therefore “in the Young” in the title (page 1, line 2)
The review article covers the issues described comprehensively, contains a lot of current literature, and is an excellent material for people dealing with echocardiography or those who want to be interested in assessing the right ventricle in children (and adults) with congenital heart disease.
RESPONSE: We thank the reviewer for the positive response, emphasizing the clinical relevance and value of our manuscript.
Some minor remarks should be made:
1.The findings of the 6th WSPH actually introduced PVR, not PVRI, into the definition of PH. These values ​​and cut-off points (3 Wood Units) should not be used interchangeably.
RESPONSE: We are grateful to the correction regarding adult PAH definition, but we follow here the recent WSPH guidelines on pediatric pulmonary hypertension (Rosenzweig et. al; Paediatric pulmonary arterial hypertension: updates on definition, classification, diagnostics and management. Eur. Respir. J. 2019, doi:10.1183/13993003.01916-2018), it is recommended to use PVR as indexed to body surface area (PVRI) in order to assess the presence of pulmonary vascular disease, as defined by PVRI ⩾3 WU·m. I hope, our definition this is in this case satisfactory to the reviewers opinion.
2. Sentence "Thus, PH or PAH with only slightly elevated PVR index is neither synonymous nor indicative of PVD in patients with CHD, e.g., systemic-to-pulmonary shunts lead to an enhanced pulmonary blood flow and consequently to an increased mPAP with or without PVD, as PVR (in children: PVR indexed [PVRi]) may be within the normal range" is not entirely clear. As far as the authors are concerned with pulmonary hypertension with the hyperkinetic flow, they should not use the term PAH, but simply PH.
RESPONSE: We agree with the reviewer and have therefore rephrased the sentence for better understanding (page 3, line 53).
- In Table 3, among the limitations of the "peak TRV" assessment, I would add the necessity to estimate the right atrial pressure for the SPAP calculation and the TRV value's dependence on the right ventricular systolic function.
RESPONSE: We added the necessity of taking into account mean right atrial pressure for estimation of sPAP, mPAP and dPAP in table 3 as well as the dependence on the RV function.
- The relationship between the pulmonary ejection acceleration time and the pulmonary artery pressure is modified by the type and location of the primary point of resistance in the pulmonary circulation. It is mentioned in the paper by A Torbicki et al. (Proximal pulmonary emboli modify the right ventricular ejection pattern. Eur Respir J. 1999 Mar; 13 (3): 616-21) and worth mentioning.
RESPONSE: We agree with the Reviewer that this fact and the respective citation is worth mentioning. We therefore included this information for the audience of Diagnostics and inserted this interesting manuscript as new Reference #60 (page 7, line 340).
- The etiology of the D-shape sign in patients with pulmonary hypertension is not entirely clear. Authors write that the left ventricle is compressed. At the same time, the invasively measured left ventricular diastolic pressure in patients with severe pulmonary hypertension is normal or even low. This indicates that the left ventricle is rather underfilled and collapsed.
RESPONSE: We agree with the opinion of this Reviewer. The LV is on one hand collapsed, underfilled and so some extent under pressure. We refer to a recent manuscript (Pulm Circ. 2019 Apr-Jun; 9(2): 2045894019854074. doi: 10.1177/2045894019854074.) (Reference #140) that includes a more detailed discussion of this topic.
Reviewer 2 Report
A very interesting publication. This is a review and does not bring new information, but it is a collection and summary of reliable knowledge about the role of individual echocardiographic parameters in the diagnosis and evaluation of patients with pulmonary hypertension. Well written, it clearly presents the principle of measuring these parameters, their usefulness and limitations. The work proves the authors' extensive knowledge and knowledge of the subject also in its practical aspect. The authors present their own very useful clinical comments and remarks on the role and importance of these parameters. The work will be very useful for everyone involved in echocadiography, cardiology and pulmonary hypertension. The information presented is supported by 174 items of current, well-chosen literature. I recommend this review for publication
Author Response
Response to Reviewer #2
A very interesting publication. This is a review and does not bring new information, but it is a collection and summary of reliable knowledge about the role of individual echocardiographic parameters in the diagnosis and evaluation of patients with pulmonary hypertension. Well written, it clearly presents the principle of measuring these parameters, their usefulness and limitations.The work proves the authors' extensive knowledge and knowledge of the subject also in its practical aspect. The authors present their own very useful clinical comments and remarks on the role and importance of these parameters. The work will be very useful for everyone involved in echocadiography, cardiology and pulmonary hypertension. The information presented is supported by 174 items of current, well-chosen literature. I recommend this review for publication.
RESPONSE: We thank the reviewer for the positive response, emphasizing the clinical relevance and value of our manuscript.
Reviewer 3 Report
The manuscript is an extensive and accurate review on the role of echocardiography in patients with congenital heart disease.
The background is correct: although international guidelines suggest several key echocardiographic variables and indices for the screening studies when PH is suspected, these may not apply to patients with congenital heart disease (CHD), due to the complex anatomy and special physiological considerations.
I have no major observations, but only minor comments to improve.
Page 4, line 98: <und> should be <and>
Pag 7, line 198, <microseconds> should be milliseconds
page 9, line 256: TAPSE is volume-dependent shouòd be changed in preload dependent
Page 11, line 282: please rephrase <When TAPSE turns abnormal in either PH patients of either age,...>
page 12, line 342: regurgitated volume should be regurgitant volume
page 14, line 395, <An LVesEI >1.0 has been shown to have prognostic value in adult with PH > is a mistake. The correct statement is < LV end-diastolic EI ³ 1.7 has been shown to have prognostic value in adult with PH ...>
page 15, line 484, has should be have
page 16, paragraph “Exercise Echocardiography”. The fact that the increase in RVSP during exercise represents an increase in RV contractility is debatable. Please refer to the study published by Ghio et al (Int J Cardiol. 2018 Nov 1;270:331-335) and to the studies cited and modify the text accordingly.
Author Response
Response to Reviewer #3
The manuscript is an extensive and accurate review on the role of echocardiography in patients with congenital heart disease.
The background is correct: although international guidelines suggest several key echocardiographic variables and indices for the screening studies when PH is suspected, these may not apply to patients with congenital heart disease (CHD), due to the complex anatomy and special physiological considerations.
I have no major observations, but only minor comments to improve.
Page 4, line 98: <und> should be <and>
RESPONSE: We agree with the reviewer and revised as requested (page 4, line 106).
Page 7, line 198, <microseconds> should be milliseconds
RESPONSE: We agree with the reviewer and revised as requested (page 7, line 206).
Page 9, line 256: TAPSE is volume-dependent should be changed in preload dependent.
RESPONSE: We agree with the reviewer and revised as requested (page 9, line 265)
Page 11, line 282: please rephrase <When TAPSE turns abnormal in either PH patients of either age,..>
RESPONSE: We agree with the reviewer that the wording of the sentence is confusing. We therefore deleted it (page 11, line 292) without loss of information.
Page 12, line 342: regurgitated volume should be regurgitant volume
RESPONSE: We agree with the reviewer and revised as requested.
Page 14, line 395, <An LVesEI >1.0 has been shown to have prognostic value in adult with PH > is a mistake. The correct statement is < LV end-diastolic EI ³ 1.7 has been shown to have prognostic value in adult with PH ...>
RESPONSE: We thank the reviewer for the thoughtful reading and revised as requested (page 14, line 414) in the main manuscript as well as in table 3.
Page 15, line 484, has should be have
RESPONSE: We agree with the reviewer and revised as requested.
Page 16, paragraph “Exercise Echocardiography”. The fact that the increase in RVSP during exercise represents an increase in RV contractility is debatable. Please refer to the study published by Ghio et al (Int J Cardiol. 2018 Nov 1;270:331-335) and to the studies cited and modify the text accordingly.
RESPONSE: We thank the reviewer for mentioning this ongoing debate. The study from Ghio et al gives important insights in dobutamine stress echocardiography data in adults with PAH. On page 16 we now discussed more detailed the study from Ghio et al, and D´Alto et al (Reference #178 and #179) providing data that confirm that the increase of PAP is only related to rest-to-exercise response in heart rate during exercise tests.